# Biocontrol Potential of *Trichoderma asperellum* CMT10 against Strawberry Root Rot Disease

**Ping Liu** [1], **Ruixian Yang** [1,*], **Zuhua Wang** [1], **Yinhao Ma** [1], **Weiguang Ren** [1], **Daowei Wei** [1] **and Wenyu Ye** [2]

[1] School of Environmental Engineering and Chemistry, Luoyang Institute of Science and Technology, Luoyang 471002, China; lylp76lp@163.com (P.L.); zuowu_zhang@163.com (Z.W.); guoqing_m@163.com (Y.M.); renweiguang02@163.com (W.R.); weidaowei05@163.com (D.W.)

[2] College of JunCao Science and Ecology (College of Carbon Neutrality), Fujian Agriculture and Forestry University, Fuzhou 350002, China; wenyuye08@163.com

\* Correspondence: fairy19790805@163.com

**Abstract:** Strawberry root rot caused by *Neopestalotiopsis clavispora* is one of the main diseases of strawberries and significantly impacts the yield and quality of strawberry fruit. Currently, the only accessible control methods are fungicide sprays, which could have an adverse effect on the consumers of the strawberries. Biological control is becoming an alternative method for the control of plant diseases to replace or decrease the application of traditional synthetic chemical fungicides. *Trichoderma* spp. are frequently used as biological agents to prevent root rot in strawberries. In order to provide highly effective biocontrol resources for controlling strawberry root rot caused by *Neopestalotiopsis clavispora*, the biocontrol mechanism, the control effects of *T. asperellum* CMT10 against strawberry root rot, and the growth-promoting effects on strawberry seedlings were investigated using plate culture, microscopy observation, and root drenching methods. The results showed that CMT10 had obvious competitive, antimycotic, and hyperparasitic effects on *N. clavispora* CMGF3. The CMT10 could quickly occupy nutritional space, and the inhibition rate of CMT10 against CMGF3 was 65.49% 7 d after co-culture. The inhibition rates of volatile metabolites and fermentation metabolites produced by CMT10 were 79.67% and 69.84% against CMGF3, respectively. The mycelium of CMT10 can act as a hyperparasite by contacting, winding, and penetrating the hyphae of CMGF3. Pot experiment showed that the biocontrol efficiency of CMT10 on strawberry root rot caused by *Neopestalotiopsis clavispora* was 63.09%. CMT10 promoted strawberry growth, plant height, root length, total fresh weight, root fresh weight, stem fresh weight, and root dry weight by 20.09%, 22.39%, 87.11%, 101.58%, 79.82%, and 72.33%, respectively. Overall, this study showed the ability of *T. asperellum* CMT10 to control strawberry root rot and its potential to be developed as a novel biocontrol agent to replace chemical fungicides for eco-friendly and sustainable agriculture.

**Keywords:** strawberry root rot; *Trichoderma asperellum*; biocontrol mechanism; biocontrol efficacy; growth-promoting effect

## 1. Introduction

Strawberries (*Fragaria ananassa*), perennial herbaceous plants belonging to the genus *Fragaria* in the Rosaceae family, are renowned for their short cultivation cycle and high economic yield. This fruit is popular among consumers because of its exceptional taste and nutritional value. Strawberries are an important economic crop both in China and globally [1]. According to data from the Food and Agriculture Organization (FAO) of the United Nations, as of 2020, China boasted a strawberry cultivation area of over 127,000 hm$^2$, with a production surpassing 3.336 million tons, ranking it as the world's leading producer of strawberries [2]. The predominant method of cultivation in China is greenhouse cultivation, which involves enclosed spaces, elevated temperatures, and high humidity. Continuous cultivation practices have led to the accumulation of pathogens, resulting

in frequent outbreaks of strawberry diseases and economic losses, which hinder the sustainable development of the strawberry industry [3]. One of the major diseases affecting strawberries is root rot, particularly in continuously cultivated fields [4]. The complex array of pathogens that contribute to strawberry root rot includes *Neopestalotiopsis clavispora*, *Phytophthora fragariae*, *Fusarium solani*, *Fusarium oxysporum*, *Rhizoctonia solani*, *Colletotrichum acutatum*, and *Armillaria mellea* [5,6]. The primary method of controlling strawberry root rot in current production practices is to use chemicals, because of the diversity of pathogens and the lack of strawberry varieties with a high resistance to root rot [7,8]. However, the use of fungicides on edible strawberry fruits poses a potential risk to human health. Hence, there is an urgent need to explore novel control strategies for strawberry root rot. Biological control measures are particularly effective to reduce soil-borne pathogens. The screening and application of biocontrol microorganisms to control root rot is very important for the sustainable development of the strawberry industry.

*Trichoderma* species have been used as biological control agents (BCAs) and as an alternative to synthetic fungicides to control a variety of plant diseases [9,10]. The biocontrol mechanisms of *Trichoderma* are based on the activation of multiple mechanisms, either indirectly, by competing for space and nutrients, promoting plant growth, plant defensive mechanisms, and antibiosis, or directly, by mycoparasitism [11,12]. Studies have indicated that *Trichoderma* spp. can increase the resistance of strawberries to root pathogens. Zhang et al. [13] found that *T. harzianum* M10-3-2 could significantly inhibit *F. solani*, which was an agent of strawberry root rot. *T. asperellum* D7-3 had remarkable growth-promoting effects on strawberries, whereas *T. koningiopsis* M0-3-3 enhanced the biocontrol efficiency of other strains against strawberry root rot. They also proved that the combination of the three *Trichoderma* strains (M10-3-2, D7-3, and M0-3-3) was more effective than individual treatments. Mercado et al. [14] discovered that *T. harzianum* could effectively control the strawberry root rot caused by *C. acutatum*. Rees et al. [15] found that *T. atrobrunneum* significantly reduces the incidence of strawberry root rot caused by *A. mellea*. Mirzaeipour et al. [16] obtained three *Trichoderma* strains with effective control against the strawberry root rot caused by *R. solani*. Despite substantial research on the use of *Trichoderma* strains to control strawberry root rot, focus has mainly been on the control of pathogens including *R. solani*, *C. acutatum*, *F. solani*, and *A. mellea*. Studies are still relatively lacking on the screening of *Trichoderma* strains against the strawberry root rot caused by *N. clavispora*.

In this study, the potential role of *T. asperellum* CMT10, isolated from healthy strawberry rhizosphere soil, was investigated as a biological control agent of the strawberry root rot caused by *N. clavispora*. To achieve this goal, plate culture, microscopy observation, and root drenching methods were employed to investigate the biocontrol mechanism and control efficiency of *T. asperellum* CMT10 on strawberry root rot, as well as its growth-promoting effect on strawberry seedlings. This study reveals that *T. asperellum* CMT10 can effectively control the occurrence of the strawberry root rot caused by *N. clavispora* and had an obvious promotion effect on strawberry seedling growth. These results indicate that *T. asperellum* CMT10 is a promising biocontrol microorganism for controlling strawberry root rot.

## 2. Materials and Methods

### 2.1. Plant Pathogen and Plant Materials

*Neopestalotiopsis clavispora* CMGF3 was isolated from strawberries with symptoms of root rot through the tissue isolation method, and the fungus was cultured on potato dextrose agar (PDA; 20% potato, 2% dextrose, 1.5% agar) for 7 d at 28 °C. *N. clavispora* CMGF3 was identified based on morphological characteristics and molecular identification. One-year-old strawberry seedlings of the commercial cultivar "Hongyan" were provided by the "Shilixiang" strawberry seedling cultivation facility.

### 2.2. Isolation and Screening of Trichoderma Strains

Soil samples were collected from the healthy strawberry rhizosphere soil of the "Shilix-iang" strawberry planting field (112°57′14.51″ E, 34°79′42.23″ N) in Luoyang, Henan Province. One gram of soil was taken in a Falcon tube (50 mL) containing sterile distilled water (SDW) and shaken (180 rpm) for 1 h. The samples were diluted from $10^{-1}$- to $10^{-5}$-fold with sterile distilled water, and a 100 µL dilution ($10^{-5}$-fold) was spread onto potato dextrose agar (PDA) plates [17]. The *Trichoderma* colonies were transferred to a new PDA medium 7 d after incubation at 28 °C for purification.

A total of 10 *Trichoderma* strains were screened for antagonistic activity against the mycelial growth of *N. clavispora* CMGF3 using a dual-culture plate assay, as described by Pimentel et al. [18]. One mycelial disc (4 mm diameter) of each *Trichoderma* sp. and *N. clavispora* was excised from the growing edges of 7-day-old cultures and placed 2 cm apart on opposite sides of PDA plates (90 mm). The plates were incubated for 7 days at 28 °C. A control of *N. clavispora* alone on PDA plates was used. The experimental design was completely randomized, with 20 treatments and three replicates. The growth rate of *N. clavispora* was determined by measuring the colony diameter. The percent inhibition was calculated as follows: percent inhibition (%) = [(pathogen colony diameter in the control treatment − pathogen colony diameter in the challenge treatment)/pathogen colony diameter in the control treatment] × 100.

### 2.3. Morphological and Molecular Identification of Trichoderma CMT10

Purified *Trichoderma* CMT10 was inoculated on a PDA plate medium and cultured in the dark for 7 days at 28 °C. Macroscopic morphology was observed, including the color and texture of the colony surface verse and reverse, the presence or absence of pigmentation, and the pattern of growth and sporulation, and images of the colonies were obtained. Microscopic morphologies such as conidia and conidiophores were observed using an optical microscope (ZEISS Axio Scope5, Oberkochen, Germany). Morphological identification relied on the descriptions found in previous research [19,20].

*Trichoderma* CMT10 was cultured in PDA medium at 28 °C for 7 days. Mycelia were harvested from the cultures, and genomic DNA (gDNA) was extracted using a DNA extraction kit (TIANGEN Biotech, Beijing, China). The extracted DNA was used as a template to amplify the internal transcribed spacer (ITS) region and the translation elongation factor-1α (*tef1-α*) region. The primers were designed with reference to previous studies [21,22]. All amplified loci, primers, and PCR conditions are presented in Table 1. PCR was performed using the TIANGEN Golden Easy PCR kit (TIANGEN Biotech, Beijing, China). The PCR products were subjected to direct automated sequencing using fluorescent terminators on an ABI 377 Prism Sequencer (Sangon Biotech, Shanghai, China). The sequences were confirmed with a BLAST (Basic Local Alignment Search Tool) search of the NCBI (National Center for Biotechnology Information) database (https://www.ncbi.nlm.nih.gov/, accessed on 23 September 2023), and a phylogenetic tree was constructed using the neighbor-joining (NJ) method, with 1000 bootstrap replications in the MEGA 10.0 package. Phylogenetic analysis with ITS-*tef1-α* gene sequences was performed to determine the position of *Trichoderma* CMT10. After identification, the sequences were submitted to Genbank. The strains used in this study and their corresponding GenBank accession numbers are listed in Table 2.

**Table 1.** Amplification sites, primer sequences, and PCR conditions used in this study.

| Gene[a] | Primer | Primer Sequence (5′-3′) | PCR Conditions | Reference |
|---------|--------|--------------------------|----------------|-----------|
| ITS | ITS1<br>ITS4 | TCCGTAGGTGAACCTGCGG<br>TCCTCCGCTTATTGATATGC | 94 °C for 5 min (94 °C for 30 s, 55 °C for 30 s and 72 °C for 40 s) × 35 cycles, 72 °C for 7 min | [21] |
| *tef-1α* | TEF-F<br>TEF-R | TGGGGCCATCAACTGAGAAAGA<br>TCTCCCTACACTTCAACTGCACA | 94 °C for 5 min (94 °C for 30 s, 53 °C for 30 s, and 72 °C for 1 min) × 35 cycles, 72 °C for 7 min | [22] |

Genes[a]: ITS, internal transcribed spacer; *tef-1α*, translation elongation factor.

**Table 2.** The ITS and *tef-1α* gene sequences of *Trichoderma* strains from the NCBI database, used for the construction of the phylogenetic tree used in this study.

| Code | Culture Accession Number(s) | Original Name | Accession no. ITS | Accession no. *tef-1α* |
|------|------|------|------|------|
| 1 | CEN1463 | *T. asperellum* | MK714888 | MK696646 |
| 2 | T34 | *T. asperellum* | LC123614 | EU077228 |
| 3 | ZJSX5002 | *T. asperellum* | JQ040324 | JQ040480 |
| 4 | KUFA0403 | *T. asperellum* | OM169354 | OP132635 |
| 5 | RM-28 | *T. asperellum* | MK092975 | MK095221 |
| 6 | TR5 | *T. longibrachiatum* | KC859426 | KC572116 |
| 7 | Tr5 | *T. harzianum* | OP938774 | OP948262 |
| 8 | DUCC001 | *T. citrinoviride* | JF700484 | JF700485 |
| 9 | S206 | *T. caerulescens* | JN715590 | JN715624 |
| 10 | TW20050 | *T. gamsii* | KU523894 | KU523895 |
| 11 | YMF1.02659 | *T. kunmingense* | KJ742800 | KJ742802 |
| 12 | CBS 121219 | *T. yunnanense* | GU198302 | GU198243 |

### 2.4. In Vitro Biocontrol of Trichoderma CMT10 against N. clavispora

2.4.1. Inhibitory Effects of Volatile Compounds

To determine the effect of the volatile compounds secreted by *Trichoderma* CMT10 against the growth of *N. clavispora* CMGF3, exposure of Trichoderma's volatile compounds was performed using the confrontation culture method [23]. Mycelial discs of Trichoderma were cut using a sterile cork borer (5 mm diameter) and were placed at the center of a freshly prepared PDA plate and cultured for 3 days at 28 °C in the dark. A mycelial disc (5 mm diameter) of the fungal pathogen *N. clavispora* CMGF3 was placed onto another freshly prepared PDA plate in the same manner. PDA plates inoculated with *N. clavispora* mycelial plugs were placed on top of the PDA plates inoculated with *Trichoderma* CMT10 and the plates were then sealed with parafilm. A control without *Trichoderma* inoculation was used and the inhibition of the mycelial growth of *N. clavispora* was observed at 28 °C for 7 d. The experiment was performed in triplicate.

2.4.2. Inhibitory Effects of Soluble Compounds

The effect of the soluble compounds of *Trichoderma* CMT10 against the growth of the fungal pathogen *N. clavispora* CMGF3 under in vitro conditions was determined as follows: *Trichoderma* CMT10 was diluted with sterile water to obtain a conidial suspension containing $1 \times 10^8$ spores/mL, and 100 μL conidial suspension was inoculated into 100 mL of potato dextrose broth (PDB) medium at 28 °C for 4 d under shaking conditions (180 rpm). The fermented liquid was centrifuged at 8000× *g* rpm for 2 min and the supernatant was filtered through a 0.22 μm filter to obtain the sterile filtrate. Therefore, the sterile filtrate was spread onto PDA plates at a ratio of 1:9 and a 7-day-old cultured *N. clavispora* CMGF3 mycelium plug was placed onto a PDA plate. A mixture of sterile water was used as the control. After 7 days of incubation at 28 °C, the diameter of the pathogen was measured and the inhibition rate (IR) was calculated. The experiment was performed in triplicate.

2.4.3. Hyperparasitism of *Trichoderma* CMT10

The hyperparasitism of *Trichoderma* CMT10 on *N. clavispora* CMGF3 was observed using a dual culture method [24]. Under sterile conditions, 1 mL of melted PDA medium was pipetted onto a sterilized glass slide to make a PDA membrane. After the solidification of the medium, *Trichoderma* CMT10 and *N. clavispora* CMGF3 mycelial discs were separately inoculated onto the membrane (with a 6 cm distance between them) at 28 °C for an incubation period of 24–72 h. Growth was recorded at 12 h intervals. After successful fungal superparasitism on the pathogen, the dual culture areas were observed using an optical microscope (ZEISS Axio Scope5, Oberkochen, Germany).

*2.5. Biochemical Properties of Trichoderma CMT10*

The precipitated $Ca_3(PO_4)_2$ on Pikovskaya's agar media (glucose, 10 g; $(NH_4)_2SO_4$, 0.5 g; NaCl, 0.3 g; $MgSO_4$, 0.3 g; $MnSO_4$, 0.03 g; $K_2SO_4$, 0.3 g; $FeSO_4$, 0.03 g; $Ca_3(PO_4)_2$, 5.0 g; agar, 15.0 g; pH 7.0–7.5) was used for the qualitative detection of the phosphate solubilizing of *Trichoderma* CMT10 [25]. Briefly, *Trichoderma* CMT10 was inoculated in Pikovskaya's agar media. The cultures were incubated for 5 days at 28 °C and the fungal growth was evaluated. Siderophore production was carried out using chrome azure S (CAS) agar media (CAS, 0.06 g; HDTMA, 0.07 g; $FeCl_3 \cdot 6H_2O$, 0.003 g; $NaH_2PO_4 \cdot 2H_2O$, 0.30 g; $Na_2HPO_4 \cdot 12H_2O$, 1.21 g; $NH_4Cl$, 0.125 g; $KH_2PO4$, 0.038 g; NaCl, 0.06 g; agar, 9.0 g; pH 6.7–6.9) [26]. *Trichoderma* CMT10 was inoculated in the CAS agar media. The cultures were incubated for 5 days at 28 °C and the fungal growth was evaluated. Nitrogen fixation was determined using nitrogen-free agar medium ($KH_2PO_4$, 0.20 g; $MgSO_4$, 0.20 g; NaCl, 0.20 g; $CaCO_3$, 5.0 g; mannitol, 10.0 g; agar, 15.0 g; pH 6.9–7.91) [27,28]. Specifically, nitrogen-free agar plates were streaked with a 5 mm disc from a PDA culture. The cultures were incubated for 5 days at 28 °C and the fungal growth was evaluated. IAA production by *Trichoderma* CMT10 was quantitatively tested according to Brick et al. [29]. A 5 mm disc was inoculated to a 100 mL Erlenmeyer flask containing 50 mL DF salts minimal medium. The medium was supplemented with 1.02 g/L tryptophan and incubated at 28 °C with continuous shaking at 150 rpm. After a 5 days incubation, 1.5 mL of culture medium was centrifuged at 10,000 rpm for 10 min. Then, a 50 µL aliquot of the supernatant was mixed vigorously with an equal volume of Salkowski's reagent in a 1.5 mL tube and incubated in the dark at 25 °C for 30 min. Uninoculated culture solution mixed with Salkowski's reagent served as a negative control and 50 µL IAA (50 mg/L) mixed with Salkowski's reagent served as a positive control. IAA production was observed as the development of a pink–red color.

*2.6. Control Effects of Trichoderma CMT10 on Strawberry Root Rot*

Mycelial discs of *Trichoderma* CMT10 and *N. clavispora* CMGF3 were inoculated at the center of PDA plates at 28 °C for 7 d. The conidial suspensions ($1 \times 10^8$ spores/mL) of *Trichoderma* and pathogen were prepared using sterile water, then stored at 4 °C for later use. One-year-old strawberry seedlings of the commercial cultivar "Hongyan" were used. The seedlings were carefully selected from the nursery with one plant per pot. Each plant was transplanted into a plastic pot (diameter, 28 cm; bottom diameter, 20 cm; height, 30 cm). Plants were grown in soil in a growth chamber at 22 °C and 75% humidity with a 16 h light/8 h dark photo period. After acclimation for 15 d, plants were used for pathogen infection and to assess the control efficiency of *Trichoderma* CMT10 on strawberry root rot. The potting root irrigation method was used for inoculation. The experiment included four treatments: (1) inoculation with *N. clavispora* CMGF3 only; (2) inoculation with *Trichoderma* CMT10 only; (3) inoculation with *N. clavispora* CMGF3 after 3 d followed by *Trichoderma* CMT10; and (4) water inoculation as a control. Each treatment consisted of 5 pots, with 3 replicates. Plants were inoculated with 5 mL of the conidial suspension of CMGF3 and CMT10 through the soil around each plant. All the treatments were followed by 60 days of incubation at 28 °C and 80% relative humidity. The disease severity of the seedlings was assessed using a scoring system of 0–5, modified from the report of Vestberg et al. [30]. Level 0 signifies an entire plant in a healthy state; Level 1 indicates a root disease incidence of ≤30%, with normal leaves; Level 2 is characterized by a root disease incidence greater than 30% and equal to or less than 60%, with normal leaves; Level 3 represents a root disease incidence greater than 60% and equal to or smaller than 80%, accompanied by yellowing leaves; Level 4 indicates a root disease incidence exceeding 80%, leading to leaf wilting; and Level 5 signifies complete plant mortality. The disease index and control efficiency were calculated based on the grading results. Disease Index = $\sum$ (disease level × number of plants at that level)/(total number of plants × highest disease level) × 100 and control efficiency (%) = (control disease index − treatment disease index)/control disease index × 100.

## 2.7. Growth-Promoting Effects of Trichoderma CMT10 on Strawberry Seedlings

The same method used in Section 2.6 was used in this experiment, which consisted of two treatments: (1) inoculation with *Trichoderma* CMT10 and (2) water inoculation as a control. Each treatment consisted of 5 pots, with 3 replicates. Plants were inoculated with 5 mL of the conidial suspension ($1 \times 10^8$ spores/mL) of CMT10 through the soil around each plant, and the plants were incubated for 60 days at 28 °C. Afterward, the strawberry seedlings were carefully excavated and their height, root length, and fresh weight (stem and leaf fresh weight, root fresh weight, and total fresh weight) were measured. The roots were dried at 45 °C in an oven, and their dry weight (g) was measured. The growth-promoting rate was calculated as follows: growth promotion rate (%) = (treatment biomass − control biomass)/control biomass × 100.

## 2.8. Data Statistics and Analysis

Data obtained from the experiments were processed using Excel 2010 and a one-way analysis of variance (ANOVA) was performed using DPS 7.05 statistical software. Duncan's new multiple range test was used to assess the significant differences, and the significance level was set at $p \leq 0.05$.

## 3. Results

### 3.1. Screening of Trichoderma Strains with Inhibitory Effects on N. clavispora

Ten *Trichoderma* strains were isolated using the dilution culture method. Two *Trichoderma* isolates, CMT10 and CMT4, were found to inhibit the mycelial growths of *N. clavispora* CMGF3, with inhibitory rates of 65.49% and 51.37%, respectively. CMT10 displayed significant inhibition activity against *N. clavispora* (Table 3). Further observations indicated that the mycelial growth of CMT10 was faster than that of CMGF3 and could, thus, quickly occupy the nutrient space. After 3 d of the dual culture, the mycelia of pathogen CMGF3 only reached one-third of the culture dish and an inhibition zone appeared between CMT10 and CMGF3. Moreover, the mycelia of CMGF3 near the inhibition zone were sparse, indicating a weakened growth. By day seven of the dual culture, the mycelia of CMT10 completely covered the CMGF3 colony and completely inhibited the growth and reproduction of CMGF3 (Figure 1). The results indicate that Trichoderma CMT10 could strongly inhibit the mycelial growth and reproduction of CMGF3, demonstrating a robust competitive advantage against the strawberry root rot pathogen.

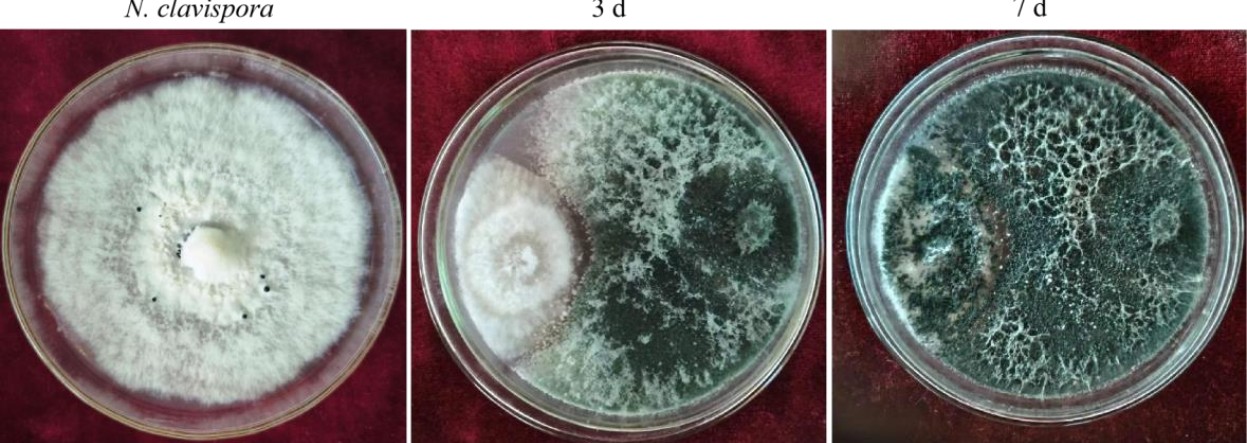

**Figure 1.** Dual cultures of *Trichoderma* CMT10 against *N. clavispora* on PDA plates.

**Table 3.** Antagonism test of *Trichoderma* strains against *N. clavispora* on PDA plates.

| Treatments | Colony Diameter (cm) | Inhibition Rate (%) |
|---|---|---|
| CMT10 | 2.93 ± 0.153 | 65.49 a |
| CMT4 | 4.13 ± 0.058 | 51.37 b |
| CMGF3 | 8.50 ± 0.000 | - |

Note: Colony diameter (cm) represented the colony diameter of *N. clavispora* CMGF3 for each treatment. Data were mean ± SD. Different letters in the same column indicated significant difference at the 0.05 level using Duncan's new multiple range test.

### 3.2. Identification of Trichoderma CMT10

*Trichoderma* CMT10 displayed a fast growth on PDA medium, with aerial mycelia completely covering the entire culture dish within three days. Initially, the colony appeared white, but it changed to yellow–green and green later. The green conidia were produced and completely covered the plate after five days (Figure 2A). Microscopically, it was observed the branches were pyramidal in type with verticillate, frequently pared lateral branches that arose from main axis with 2–5 phialides clustered at the top. The angle with the main axis was 90°, and the lateral branches re-branched. The phialides were ampulliform, somewhat thicker in the middle, and terminated with conidia (Figure 2B). The conidia were spherical to ellipsoidal, 2 (−3.7) × 3.2 (−4.5) μm, single-celled, and light green (Figure 2C). Based on these cultural and morphological characteristics, the strain CMT10 was identified as *T. asperellum*.

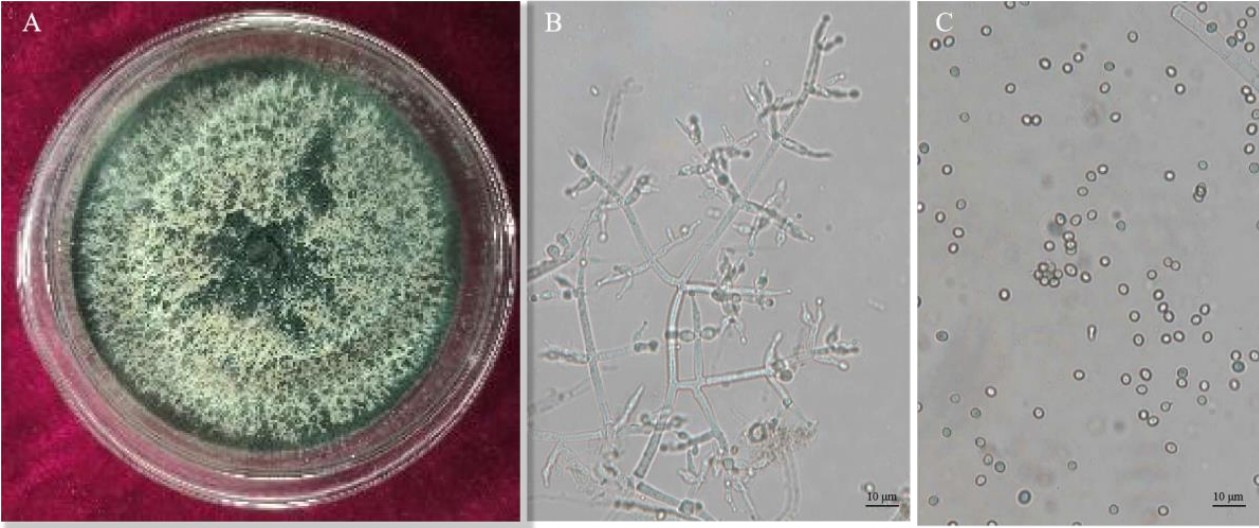

**Figure 2.** Cultural and morphological characteristics of *Trichoderma* CMT10. (**A**) The colony morphologies on PDA medium incubated at 28 °C for 7 days; (**B**) conidiophores; and (**C**) conidia. Scale bar = 10 μm.

The ITS regions and *tef1-α* regions of *Trichoderma* CMT10 were amplified and sequenced. The GenBank accession numbers were PP126513 and PP171486, respectively. The phylogenetic tree based on the ITS-*tef1-α* gene sequences showed that *Trichoderma* CMT10 was closely related to *T. asperellum* strains CEN1463, T34, ZJSX5002, KUFA0403, and RM-28 (Figure 3). The details of the strain names, origins, and accession numbers are listed in Table 2. Therefore, the CMT10 strain was identified as *T. asperellum*, according to morphological characterization and molecular analysis.

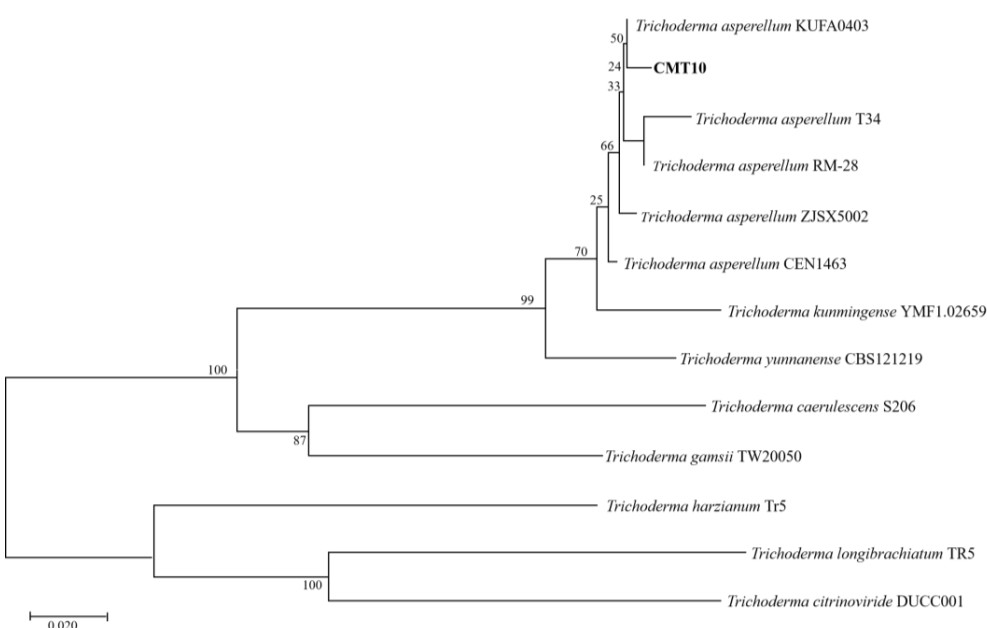

**Figure 3.** The phylogenetic tree of *Trichoderma* CMT10, based on the ITS-*tef1-α* gene sequences and their homologous sequences. Phylogenetic trees were constructed using the neighbor-joining method of MEGA 10.0 with bootstrap values based on 1000 replications. The accession numbers of the sequences are provided in Table 2. Bootstrap values are shown at branch points. The scale bar indicates 0.020 substitutions per nucleotide position.

### 3.3. In Vitro Biocontrol of Trichoderma CMT10 against N. clavispora

3.3.1. Inhibition Rates of Volatile Metabolites from *Trichoderma* CMT10 on *N. clavispora*

The effect of volatile metabolites emitted by *T. asperellum* CMT10 was tested against the growth of *N. clavispora* using the confrontation culture method. The mycelia of pathogenic CMGF3 were inhibited significantly by the volatile metabolites of CMT10, compared to the control. The IR was 69.84% at 7 d after confrontation culture (Figure 4). On the tenth day, the mycelia of the pathogenic CMGF3 had ceased to grow, while the mycelia of *T. asperellum* CMT10 continued to spread and encroach upon the colony of the pathogenic CMGF3.

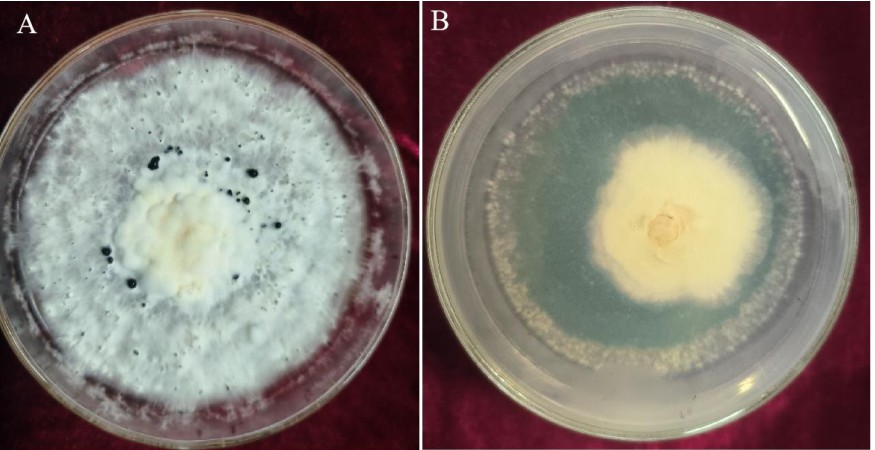

**Figure 4.** The inhibitory effect of volatile metabolites produced by *T. asperellum* CMT10 against the growth of the fungal pathogen *N. clavispora*. (**A**) PDA plate inoculated with *N. clavispora* and (**B**) the PDA plate inoculated with *N. clavispora* was placed on top of the PDA plate inoculated with *T. asperellum* CMT10 for 7 d at 28 °C, and the colony diameter was measured.

### 3.3.2. Inhibition Rates of Soluble Metabolites from *Trichoderma* CMT10 on *N. clavispora*

The antifungal activity of soluble metabolites produced by *T. asperellum* CMT10 was assessed against the fungal pathogen CMGF3. The results demonstrated that the soluble metabolites of CMT10 had a strong inhibitory effect against the growth of the fungal pathogen CMGF3 on PDA plates. After 7 d of incubation at 28 °C, the colony diameter of the CMT10-treated fungal growth was reduced significantly, compared to the untreated control (Figure 5). The IR of soluble metabolites produced by *T. asperellum* CMT10 was 79.67% against CMGF3.

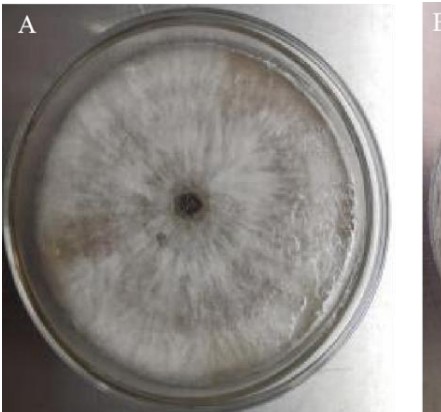
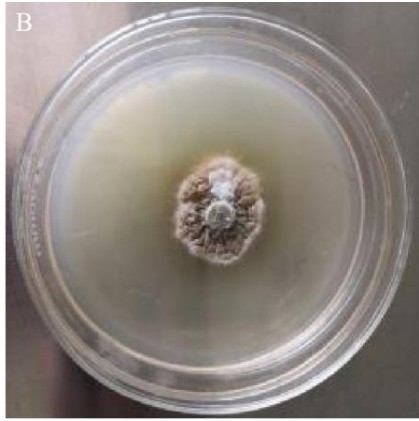

**Figure 5.** The inhibitory effect of soluble metabolites produced by *T. asperellum* CMT10 against the growth of the fungal pathogen *N. clavispora*. (**A**) inoculated with *N. clavispora* on the PDA plate mixed with sterile water and (**B**) inoculated with *N. clavispora* on the PDA plate mixed with the sterile filtrate of *T. asperellum* CMT10 for 7 d at 28 °C, and the colony diameter was measured.

### 3.3.3. Hyperparasitism of *Trichoderma* CMT10 on *N. clavispora*

Microscopic observation of the hyphal interaction between *T. asperellum* CMT10 and the pathogen CMGF3 revealed that the mycelia of both strains began to contact each other, but the antagonistic effect between them was not evident after 48 h (Figure 6A). CMT10 mycelia were attached to CMGF3 after 72 h (Figure 6B). After 96 h, CMT10 mycelia grew along and entwined with CMGF3 mycelia, causing the contraction of CMGF3 mycelia (Figure 6C,D). Moreover, CMGF3 mycelia were observed being penetrated and were embedded by CMT10 mycelia (Figure 6E). The results showed that *T. asperellum* CMT10 indicates a strong hyperparasitic effect against the strawberry root rot pathogen *N. clavispora* CMGF3.

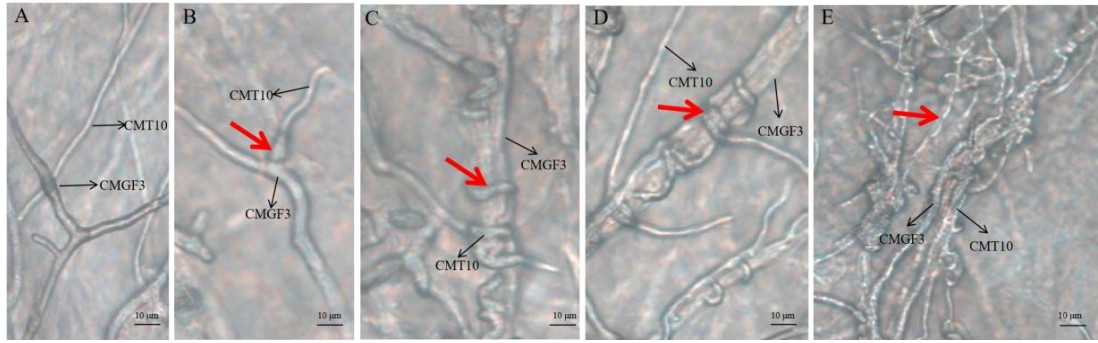

**Figure 6.** The hyperparasitic effects of *T. asperellum* CMT10 against *N. clavispora*. (**A**) Healthy mycelium morphology of CMT10 and healthy mycelium morphology of CMGF3; (**B**) the hyphae of CMGF3 were attached by CMT10 hyphae (as shown by the red arrow); (**C,D**) the hyphae of CMGF3 were entangled by CMT10 hyphae (as shown by the red arrow); and (**E**) the hyphae of CMGF3 were penetrated by CMT10 hyphae (as shown by the red arrow). The hyphae of CMT10 and CMG3 were determined primarily by the difference in mycelia diameter under microscopic conditions.

*3.4. Determination of Biochemical Properties of T. asperellum CMT10*

The results of the biochemical properties' determination revealed that the mixed solution of *T. asperellum* CMT10 with the Salkowski reagent did not turn pink, indicating that CMT10 could not produce indole-3-acetic acid (IAA), but it could grow on inorganic phosphate medium (Figure 7A). In the nitrogen-fixing medium, the mycelia were sparse, sporulation was limited, and spore distribution exhibited a spotty pattern (Figure 7B). Moreover, CMT10 grew on siderophore medium, demonstrating its ability to produce siderophores (Figure 7C).

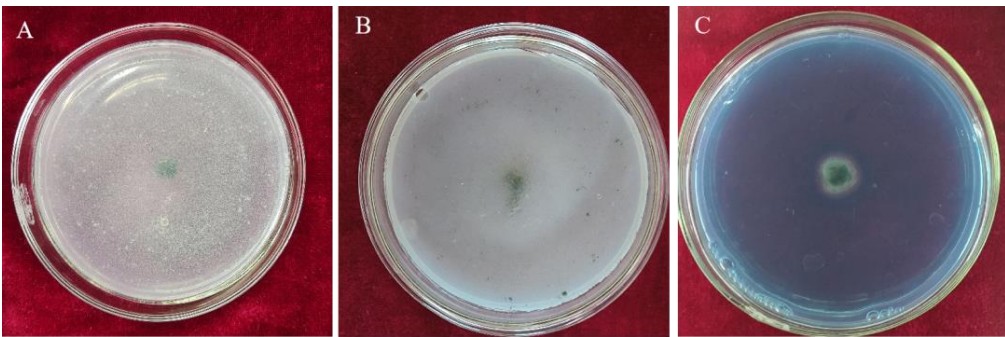

**Figure 7.** The biochemical properties of *T. asperellum* CMT10. (**A**) The ability of inorganic phosphorus solution; (**B**) the ability of nitrogen fixation solution; and (**C**) the ability of siderophore production.

*3.5. Biocontrol Efficiency of T. asperellum CMT10 against Strawberry Root Rot*

The incidence of strawberry root rot on each treatment was investigated after inoculation for 60 days (Table 4, Figure 8). The results revealed that the inoculation with *T. asperellum* CMT10 and the water control did not exhibit disease symptoms in strawberries. Treatment with the inoculation of *N. clavispora* CMGF3 showed the most severe disease symptoms, with a disease index of 84.00, which was significantly higher than that of the other treatments ($p \leq 0.05$). The disease index for treatment with *N. clavispora* CMGF3 + *T. asperellum* CMT10 was 31.00 and its biocontrol efficiency against strawberry root rot reached 63.09%, indicating that *T. asperellum* CMT10 effectively controlled the occurrence of potted strawberry root rot.

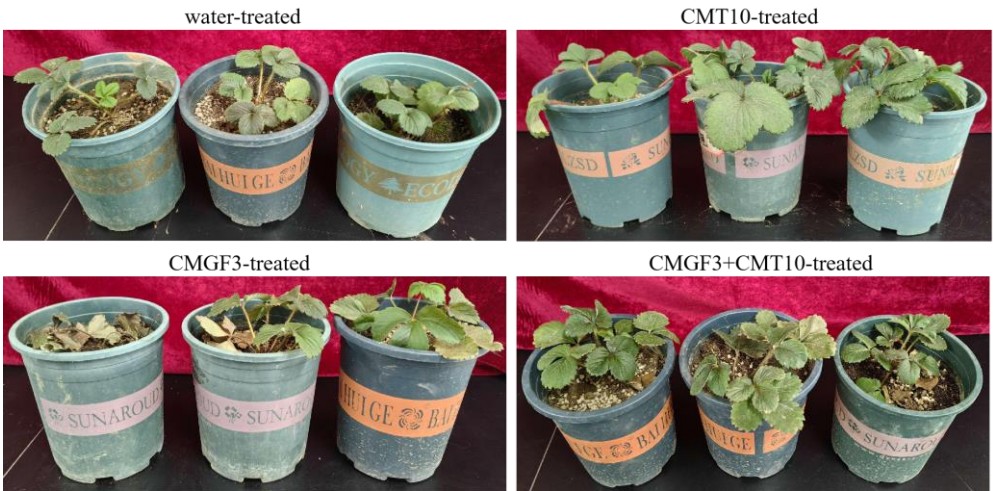

**Figure 8.** The control efficiency of *T. asperellum* CMT10 against strawberry root rot.

**Table 4.** The control efficiency of *T. asperellum* CMT10 against strawberry root rot.

| Treatments | Disease Index | Control Efficiency (%) |
|---|---|---|
| CMGF3 | 84.00 ± 0.04 a | - |
| CMT10 | 0.00 ± 0.00 c | - |
| CMGF3 + CMT10 | 31.00 ± 0.61 b | 63.09 ± 0.07 a |
| Control | 0.00 ± 0.00 c | - |

Note: water inoculation was used as a control. Data were mean ± SD. Different letters in the same column indicate a significant difference at the 0.05 level using Duncan's new multiple range test.

### 3.6. Growth-Promoting Efficiency of T. asperellum CMT10 on Strawberry Seedlings

The growth-promoting effects of *T. asperellum* CMT10 on strawberry seedlings were investigated after 60 days of inoculation. There was a significant increase in plant height, root length, total fresh weight, root fresh weight, stem fresh weight, and root dry weight compared with the water control. The growth-promoting rates were 20.09%, 22.39%, 87.11%, 101.58%, 79.82%, and 72.33%, respectively (Table 5, Figures 9 and 10).

**Table 5.** Growth-promoting effects of *T. asperellum* CMT10 on strawberry seedings.

| Treatments | Plant Height (cm) | Root Length (cm) | Total Fresh Weight (g) | Root Fresh Weight (g) | Stem Fresh Weight (g) | Root Dry Weight (g) |
|---|---|---|---|---|---|---|
| CMT10 | 12.57 ± 1.35 a | 23.75 ± 2.18 a | 13.55 ± 3.53 a | 7.18 ± 3.37 a | 6.37 ± 2.08 a | 2.66 ± 1.00 a |
| Control | 10.53 ± 1.41 b | 19.67 ± 2.70 b | 7.61 ± 1.66 b | 3.87 ± 1.59 b | 3.74 ± 0.61 b | 1.56 ± 0.50 b |

Note: water inoculation was used as a control. Data were mean ± SD. Different letters in the same column indicate a significant difference at the 0.05 level using Duncan's new multiple range test.

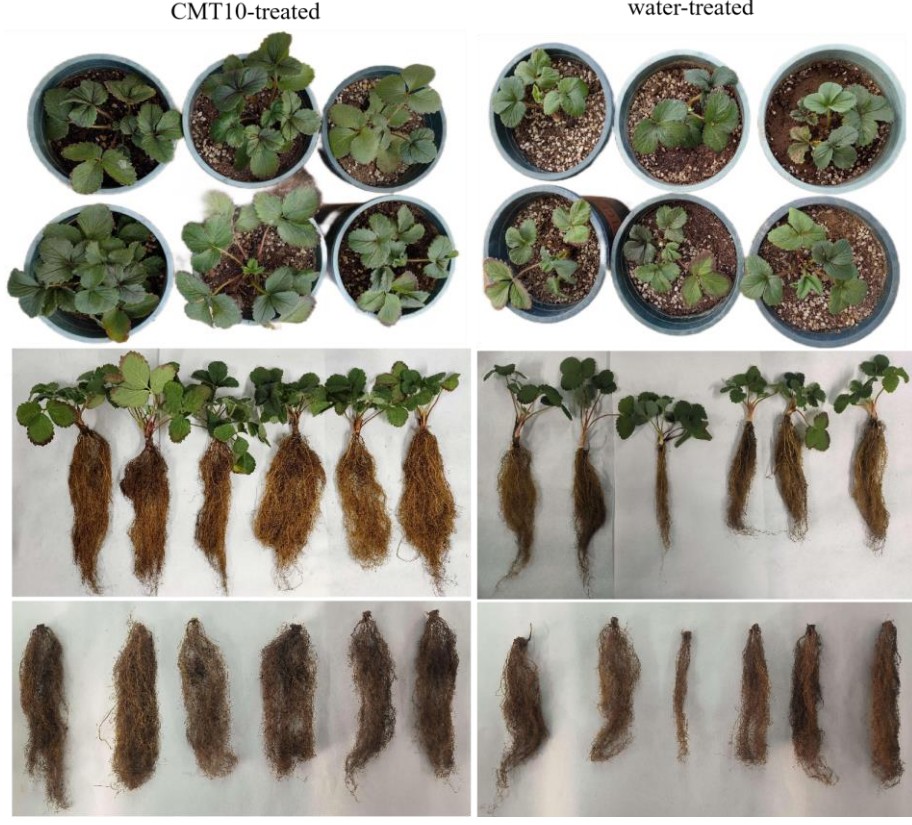

**Figure 9.** Growth-promoting effects of *T. asperellum* CMT10 on strawberry seedlings.

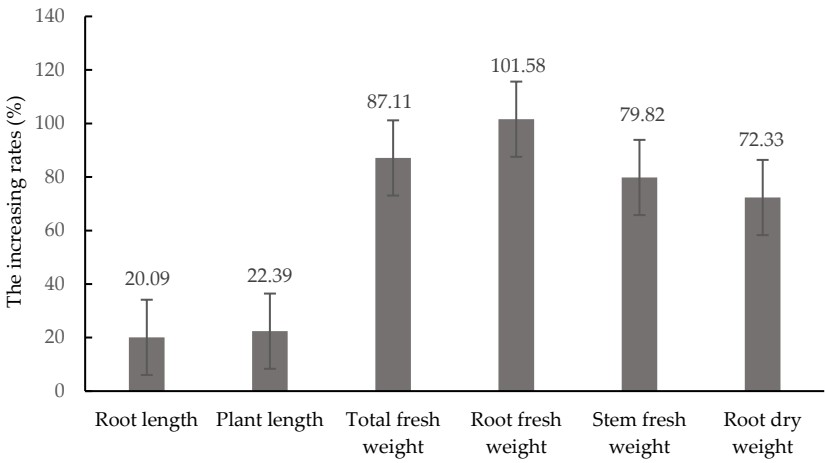

**Figure 10.** The increasing rates of *T. asperellum* CMT10 on the biomass of strawberry seedlings.

## 4. Discussion

### 4.1. Significance of Exploring Biocontrol Resources for Strawberry Root Rot

The prevention of strawberry root rot is complicated due to its diverse composition of pathogens, making it difficult to control. This disease is responsible for significant economic losses to the strawberry industry under greenhouse cultivation [31]. *Trichoderma* spp., recognized as crucial biocontrol resources, have been widely utilized in the disease control of various crops [32]. *Trichoderma* spp. have played a pivotal role in the prevention and control of strawberry root rot. However, due to the diverse composition of root rot pathogens, most studies have focused on *Trichoderma* against *Fusarium* spp. and *Rhizoctonia solani* [33,34]. Studies focusing on *N. clavispora*, a pathogen associated with strawberry root rot, are scarce. In this study, *N. clavispora*, an important pathogen causing strawberry root rot, was specifically selected as a target pathogenic fungus and obtained a strain of *T. asperellum* CMT10 with significant inhibition activity against *N. clavispora*. This study highlighted the remarkable effectiveness of *T. asperellum* against strawberry root rot and its ability to promote growth in strawberry seedlings. These findings contributed to the development of biocontrol resources for managing strawberry root rot and broadened the potential applications of *T. asperellum*.

### 4.2. Biocontrol Mechanism of T. asperellum CMT10

Most studies had demonstrated that *Trichoderma* strains could inhibit pathogenic fungi through nutrient and spatial competition, hyperparasitism, and the production of antimycotic secondary metabolites, while they could also promote plant growth and enhance plant stress resistance [35]. Ecological niche competition is a crucial mechanism of biocontrol microorganisms for preventing disease in biocontrol. *Trichoderma*, a biocontrol agent, is able to rapidly occupy ecological niches in environments with low concentrations of nutrients, which can cause pathogenic fungi to lose their ability to thrive and survive [36]. Risoli et al. [37] found that the growth rate of *Trichoderma* was 2.0–4.2 times that of *Botrytis cinerea*, indicating a significantly faster growth of *Trichoderma* compared to the pathogen, impeding the growth and reproduction of the pathogen. The results of this study indicated that *T. asperellum* CMT10 could significantly inhibit the growth and reproduction of *N. clavispora*. In the initial phase, CMT10 exhibited rapid growth and a strong competitiveness, and it quickly occupied nutritional and ecological spatial sites and produced an inhibition zone. In the later stages of cultivation, the *N. clavispora* colony completely disappeared and was replaced by dark green conidia of *T. asperellum*.

*Trichoderma* employs the mechanism of antibiosis in its biological control. Metabolites produced by *Trichoderma*, both volatile and soluble, have been reported to restrict the growth of various pathogenic fungi [38]. The metabolites included triohodexrmin, gliotoxin, viridin, and peptide antimycotics [39]. Naglot et al. [40] found that metabolites of *Trichoderma*

significantly inhibited *F. oxysporum* with an inhibition rate of up to 54.81%. Manganiello et al. [41] discovered that volatile secondary metabolites secreted by *T. viride* TG050609 caused irregular growth, fragmentation, and even dissolution of *Phytophthora nicotianae*. By determining the inhibitory effects of the soluble and volatile metabolites of *T. asperellum* CMT10 on the *N. clavispora,* causing strawberry root rot, it was found that after 7 days of cultivation on CMT10-fermented metabolite plates, the inhibition rate reached 79.67% and the inhibition rate of volatile metabolites against *N. clavispora* reached 69.84%. This suggests that CMT10 metabolites have a strong inhibitory effect on the *N. clavispora* that causes strawberry root rot. However, the metabolites responsible for this effect have not been identified in our study and require further investigation.

Hyperparasitism was a vital mechanism employed by *Trichoderma* for its biological control. *Trichoderma* recognized lectins on the mycelia of a pathogenic fungi and engages in processes such as identification, contact, wrapping, penetration, parasitism, and dissolution of the fungi [42]. Hewedy et al. [43] found that *T. harzianum* Th6 could adhere to, invade, and disrupt the mycelia of *F. graminearum*. Larran et al. [44] observed that *T. harzianum* could form adhesive structures on the mycelia of *F. sudanense*, leading to curling, wrinkling, and dissolution of the mycelia of *F. sudanense*. The present study also found that *T. asperellum* CMT10 exhibited hyperparasitism against *N. clavispora*. It could recognize, contact, wrap, and parasitize the mycelia of the pathogen. However, mycelium dissolution, protoplasm leakage, or cell disintegration were not observed, which may be related to the observation time during cultivation. It is believed that the cell wall hydrolytic enzymes secreted by *Trichoderma* played a crucial role in its hyperparasitic activity, such as chitinases, glucanases, and proteases, which can dissolve the cell walls of pathogenic fungi, allowing *Trichoderma* to parasitize, absorb nutrients, and ultimately cause the death of the pathogenic fungi [45]. Whether *T. asperellum* CMT10 can secrete enzymes with lytic effects was not investigated in our study and is a direction for future research.

### 4.3. Practical Application of T. asperellum CMT10

Currently, the production of *Trichoderma* generally involves the simultaneous or sequential action of several disease prevention mechanisms. *Trichoderma* can utilize different antagonistic mechanisms at different stages to biocontrol effects [46]. This study demonstrated that *T. asperellum* CMT10 exerted competitive, antibiosis, and hyperparasitic effects against the pathogenic fungus causing strawberry root rot. *T. asperellum* CMT10 could effectively control the occurrence of strawberry root rot. However, the biocontrol mechanisms at different stages of interaction between *Trichoderma* and the pathogenic fungi in plants still need further exploration, which may provide a theoretical foundation for the practical application of *T. asperellum* CMT10 in strawberry production. Therefore, future research should focus on the field disease control effect and the interactive relationships among *T. asperellum* CMT10, the pathogenic fungi causing root rot, and the host plant. In addition, this study specifically evaluated the growth-promoting effects of *T. asperellum* CMT10 on strawberry seedlings by measuring parameters such as plant height, root length, total fresh weight, root fresh weight, stem fresh weight, and root dry weight. It is essential to conduct more field experiments to fully understand the growth-promoting effects of CMT10 on strawberry plants, as well as to investigate the impact of *T. asperellum* CMT10 on strawberry fruit size, yield, and quality.

### 5. Conclusions

In summary, *T. asperellum* CMT10 was obtained among 10 *Trichoderma* strains as a potent biocontrol agent against *N. clavispora*, the pathogenic fungi causing strawberry root rot. The results of the pot experiment demonstrated that *T. asperellum* CMT10 effectively inhibited root rot and significantly enhanced the growth of strawberry seedlings. These findings indicate that *T. asperellum* CMT10 has great potential as a biocontrol resource for preventing and controlling strawberry root rot, making it a promising candidate for future development.

**Author Contributions:** Conceptualization, R.Y.; methodology, P.L., Z.W., W.R. and R.Y.; software, D.W., Y.M., W.Y. and W.R.; formal analysis, P.L., W.R. and R.Y.; writing—original draft preparation, R.Y.; visualization, D.W., Y.M. and W.R.; project administration, R.Y. and W.Y.; funding acquisition, W.Y. and R.Y. All authors have read and agreed to the published version of the manuscript.

**Funding:** This research was funded by the Science and Technology Program of Henan province of China (232102320111), the Science and Technology Planning Major Project of Fujian province of China (2022N0010), and the Natural Resources Science and Technology Innovation Project of Fujian province of China (KY-090000-04-2022-016).

**Data Availability Statement:** The original contributions presented in the study are included in the article, further inquiries can be directed to the corresponding author.

**Conflicts of Interest:** The authors declare no conflicts of interest.

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
