# Peer review of "Biocontrol Potential of Trichoderma asperellum CMT10 against Strawberry Root Rot Disease"

_horticulturae, doi:10.3390/horticulturae10030246_

Round 1
Reviewer 1 Report
Comments and Suggestions for Authors
Abstract : The abstract provides a concise overview of the study on the biocontrol potential of Trichoderma asperellum CMT10 against strawberry root rot caused by Neopestalotiopsis clavispora. However, it could be improved by reorganizing some information for better clarity and flow.
Introduction : The introduction effectively contextualizes the significance of strawberry cultivation and the challenges posed by root rot diseases. However, it could benefit from streamlining and organizing some information for improved readability.
Materials and Methods : The Materials and Methods section is detailed and provides a clear outline of the experimental procedures. However, it could be further improved by organizing the information into subsections for better clarity. Here's a suggested structure:
Plant Pathogen and Plant Materials
Isolation and Screening of Trichoderma Strains
Morphological and Molecular Identification of Trichoderma CMT10
Biocontrol Mechanism of Trichoderma CMT10 against N. clavispora 4.1. Inhibitory Effects of Volatile Compounds 4.2. Inhibitory Effects of Non-volatile Compounds 4.3. Hyperparasitism of Trichoderma CMT10
Growth Promotion Properties of Trichoderma CMT10
Control Effects of Trichoderma CMT10 on Strawberry Root Rot
Growth-Promoting Effects of Trichoderma CMT10 on Strawberry Seedlings
Data Statistics and Analysis
Overall, the Results and Discussion section of the scientific article provides a comprehensive overview of the experimental findings regarding the screening, identification, biocontrol mechanisms, and practical application of Trichoderma asperellum CMT10 against Neocosmospora clavispora, the pathogen causing strawberry root rot. Here are some suggestions to improve the quality of the manuscript:
Clarity and Organization:
Consider restructuring the section to improve flow and coherence. The current organization seems slightly disjointed, with abrupt transitions between subsections. Try to establish a smoother transition between each subsection to enhance readability.
Start each subsection with a clear introductory statement that summarizes the main findings or objectives to provide context for the reader.
Data Presentation:
Ensure consistency in presenting data. For example, Table 3 and Figure 4 present similar information about inhibition rates, but the data are presented differently. It may be helpful to standardize the format for better clarity.
Provide error bars or confidence intervals for measured values in tables and figures to convey the variability of the data.
Figures and Tables :
Figures and tables should be clearly labeled and referenced in the text. Ensure that each figure and table is mentioned in the Results section and discussed in the corresponding subsection.
Consider providing more detailed captions for figures to explain key observations or results directly, without the need for the reader to refer back to the text.
Discussion :
Expand on the implications of the findings and their significance for the field of plant pathology and biocontrol. Discuss how the results contribute to existing knowledge and potential applications in agricultural practices.
Provide a more critical analysis of the limitations of the study and areas for future research. Identify any potential biases or confounding factors that may have influenced the results.
Conclusions :
Summarize the key findings of the study concisely in the conclusion section. Highlight the main contributions of the research and its implications for future research or practical applications.
Avoid introducing new information or data in the conclusion section. It should serve as a synthesis of the results presented earlier in the manuscript.
Comments on the Quality of English LanguageLanguage and Grammar :
Review the text for grammatical errors, awkward phrasing, and typos. Ensure that the language is concise and precise to convey the findings effectively.
Avoid repetition of phrases or terms. For example, "inhibition rate" is mentioned multiple times. Consider using synonyms or rephrasing to improve readability.
Reviewer 2 Report
Comments and Suggestions for Authors
Dear Authors,
I have reviewed your manuscript "Biocontrol potential of Trichoderma asperellum CMT10 against strawberry root rot disease", submitted for publication in Horticulturae.
After reading your manuscript, I can tell that your research is well-planned, the experimental design is appropriate, scientifically sound and correctly carried out, the experiments are clearly presented and properly discussed. The language of the manuscript is somewhat flawed, and thus most of my comments and remarks are focused on language, although occasionally I will ask you for some clarifications, elaborations, or logical corrections. Please follow my recommendations for revision as given below:
· Abstract:
o line 11-15: "There are now no effective control techniques available except for fungicide sprays" and "Trichoderma is widely used as a biological agent for controlling strawberry root rot" – these two sentences are in obvious contradiction, please revise
o line 20 and elsewhere in the manuscript: Please replace the word "antibiotic" with "antimycotic" throughout the manuscript text. The modern meaning of "antibiotic" is considered synonymous with "bactericidal" whereas you want to kill a fungal pathogen
o line 24: please delete the word "making"
o line 28: please replace "were" with "being"
o line 30: please put this sentence into singular: "...and its potential to develop as a novel biocontrol agent..."
· Introduction:
o line 37: please put Fragaria into italic letters
o line 39: "globally and in China", not "in globally and China"
o line 53: high resistance TO root rot
o line 54: poses, not posed (this is still happening)
o line 64: please add the word "to": "of strawberry TO root pathogens"
o line 66-68: please stick to one single tense over this entire sentence (present tense should be preferred over the past tense)
o line 84-87: The last sentence of the Introduction is completely redundant with the previous sentence, you are repeating the same things again. You should delete this sentence, and consider replacing it with a sentence in which you would give the readers a hint of your current findings, that T. asperellum CMT10 can be efficiently used for biocontrol of the N. clavispora pathogen.
· Materials & Methods:
o line 91: why "our preliminary work"? You are literally reporting on that in your current work. so it is not preliminary. Instead, you should give just a brief justification on how you identified the pathogenic strain as N. clavispora CMGF3.
o line 97: the geolocalization coordinates seem weird. There are 60 minutes (') in one degree (°) and 60 seconds ('') in one minute ('), so 79', 1451'', and 4223'' are numbers that look unlikely. Please double-check whether the geolocalization coordinates are given in minutes and seconds, or in decimal numbers, and then revise, accordingly.
o line 99: Diluted how? In which proportion?
o line 200: The title of 2.5. should be revised. I suggest "Biochemical properties..." or something similar. Although these biochemical properties indeed act as plant growth-promoting, there is later another section (2.7) in which you talk again about plant growth-promoting properties, and in that section you are actually talking about direct effects on seedlings in vivo.
o Another very important remark is that in this section (2.5) you should briefly elaborate how each of these three biochemical assays actually work. What is the principle of each of these assays, what do they measure, and how are these results interpreted? Nitrogen fixation can be inferred from the context, phosphorus solubilization is somewhat clear but not entirely, but the readers will not be able to grasp the meaning of the siderophore production assay. Is it for solubilization of potassium? Or iron? Or both? What does it measure and how? Provide a brief elaboration for each of the three of them within this section. It is good that you provided the references for each of them (please do keep the references!), but the readers will appreciate getting briefly acquainted with each of them without having to download the original papers to look for general principles of how these methods work.
o line 211: please replace " to biosynthesis IAA" with "FOR biosynthesis OF IAA"
· Results:
o Figure 3: the branching nodes next to the part of the phylogenetic tree where T. asperellum strains are shown, are too close to the end of the diagram, so the branching is not readily visible. However, this is exactly the most important part of the phylogenetic tree. I suggest that the length of the branches (horizontal lines) in the diagram should be changed, so that all the branching nodes are moved left from where they currently are. This will make the branching of the diagram in the part corresponding to T. asperellum strains much better visible.
o line 308: The title of the section 3.3 should be changed. Although you demonstrate different pathways of Trichoderma-mediated biocontrol (volatile compounds, soluble compounds, hyperparasitism on hyphae), you did not really show the biocontrol mechanism for most of these pathways. The word "mechanism" should thus be deleted from the title. I suggest "In vitro biocontrol of Trichoderma CMT10 against N. clavispora" because these are all in vitro assays, and later, you are going to demonstrate the biocontrol effects in vivo.
o line 318 and further: at many places throughout the manuscript, "clavispora" has been changed to "clavisporain". Please revise throughout the manuscript text.
o line 322 and elsewhere: I would suggest to replace the wording "non-volatile" with "soluble", both here, and throughout the manuscript text. What you obtained by this experiment, was the soluble fraction of Trichoderma's metabolites that exert a biocontrol effect. I understand that you used the word "non-volatile" as opposed to the volatile metabolites in your other experiment; however, we cannot be entirely sure that all of these metabolites, that are excreted into the medium, are indeed "non-volatile" – some of them may be volatile to some extent, but they still end up being excreted into the growth medium. For this reason I would suggest replacing the word "non-volatile" with "soluble" throughout the manuscript text.
o section 3.3.3: the phenomenon observed in Figure 6 is extremely interesting, and what you show in Figure 6 is indeed compelling. Also, since you observed an obvious suppressive effect of Trichoderma on N. clavispora (and not vice-versa) it is intuitively obvious to the observer of Figure 6, that the "aggressor hyphae" are those of Trichoderma, and those that are "being aggressed" are those of N. clavispora. However – is there another, reliable criterium (other than intuitive common sense) by which you could actually differentiate between the hyphae of the two species, and thus convincibly demonstrate that this is Trichoderma parasitizing the hyphae of N. clavispora, and not vice-versa? If not, that is still alright but in that case you should make it clear that you are interpreting the micrograph based on common sense rather than based on a exact criterium. In each case, please elaborate within the manuscript text, or at least within the figure caption.
o caption to Figure 6: "hyphae" is plural of "hypha". Thus, you cannot say, "hyphae was", you should always say "hyphae were". Also, you cannot say "hyphaes".
o sections 3.4, 3.5, and 3.6: these sections are incorrectly numbered as "2.4", "2.5", and "2.6", please revise
o line 353: How did you determine that Trichoderma was unable to produce IAA? Please briefly explain, and also add the methodology to the Materials & Methods section.
o caption to Figure 7: This caption is very poorly worded. The nutrient solutions do not have any ability; it is Trichoderma that has various abilities that are demonstrated through the corresponding assays. Please extensively revise, and explain thoroughly what it is that we are seeing in the pictures. Since you will revise and amend the methodology section 2.5 as well, you may find inspiration for this figure caption in the revise text of the methodology.
o line 363: please replace "...that the treatment with inoculation of" with just "...that the inoculation with"
o line 366: P should be written in italic. Also, in line 256 you used lowercase p for confidence interval. You can choose either uppercase P or lowercase p but please always use consistently just one of them, and make sure that it is written in italic.
o Table 3 and elsewhere – efficiency/efficacy: In table 3, you are using the word "efficiency", although throughout the manuscript you are using the word "efficacy" much more often. Please pick one of these two terms and always stick to it consistently (personally I would prefer "efficiency" but you may use either of them).
o Table 3: you say that control efficiency/efficacy is 63.00%, but according to your own formula my calculation says 63.09%. Please double-check and revise, if necessary.
· Discussion:
o line 393-395, 433-434: The grammar of these two sentences is messed up in multiple ways. Please extensively revise.
o line 403: antimycotic, promotE
o throughout the Discussion, many sentences are written in past tense for no reason. Please put the following into present tense: line 404 ("was"), 406 ("was"), 416 ("employed"), 430 ("was").
o line 428-429: please replace "remain unclear" with "have not been identified in our study"
o line 445: please replace "remains unclear" with "was not investigated in our study"
o line 457: please put CMT10 into plain text
o line 466: please replace "the pathogenic fungi caused" with "the pathogenic fungUS causING"
I look forward to reading your published paper in Horticulturae.

The language of the manuscript is somewhat flawed, but mostly in terms of grammar (rather than in terms of scientific terminology), which can be improved through a language editing service. I have pointed to most of the points for English revision but the grammar should be thoroughly revised anyway. Please also make sure not to use the past tense for general phenomena (such as, the mode of action of Trichoderma, and similar).
Reviewer 3 Report
Comments and Suggestions for Authors
In this manuscript, the authors selected T. asperellum strain CMT10, which showed the greatest antibiotic effect on Neopestalotiopsis clavispora, the causal agent of strawberry root rot. Then they conducted numerous experiments in which they demonstrated the usefulness of T. asperellum CMT10 as biological control agents. They demonstrated not only the degree of reduction in the growth of N. clavispora, but also the mechanism of antagonistic effects, including antibiosis (volatile and non-volatile compounds) and mycoparasitism (demonstrated by microscopical observations). The antagonistic properties of T. asperellum CMT10 were also confirmed in in vivo tests, which is a very important element. Moreover, they proved the growth-promoting effects of this fungal strain on strawberry. This is an original and very valuable achievement. The introduction presents the research problem in a comprehensive way and ends with defining the purpose of the research. The methods are described in sufficient detail. Numerous experiments are well documented. Discussion is interesting. The manuscript should be published in Horticulturae. However, there are numerous topographical and other small errors, for example those indicated in Remarks. In some places the authors do not pay attention to whether Sigular or Plural should be used. Therefore, the manuscript requires minor changes before publication.
Remarks
Lines 14, 59, 61, 69, 74, 77: Trichoderma - it should be in italic, it should be corrected throughout the manuscript
Lines 17, 69, 167, 168, 174, 193, 243, 250, 286, 287, 313, 358: consider revising these texts, there are typographical errors
Line 20 hyperparasitism effect – should be hyperparasitic effect
Line 37 Fragaria - it should be in italic
Line 40 China[1]. – there should be a space, this applies to the entire manuscript
Line 79-83 I suggest removing this fragment because the same thing is repeated in lines 84-87 as the purpose of the work
Line 118 conidiophore or conidiophores ?
Line 207 nitrogenfree – rather nitrogen-free?
Line 249 dry weight - enter units
Line 272 of ?
Line 298 Conidiophore – rather Conidiophores
Line 271 Figure 1 – this should be higher in Line 272 !!!
Table 3 ‘Colony diameter’ – specify which fungus it concerns. The current situation indicates that 2.93±0.153 is the diameter of CMT10 ...
Table 3 colony diameter - add units, cm?
Line 274 ‘The same below.’ – not clear
Line 276, Line 318, Line 334 'N. clavisporain’ ?? – wrong fungal name
Line 324 ‘The antibacterial activity’ – this is an error, you study antifungal activity
Table 4 CK - needs explanation
Line 333 T. asperellum – it should be in italic, this aspect should be corrected for all Latin names of fungal species and genera in text and in Referencers
Line 348 349, 350 – please note that ‘The hyphae’ is Plural – so ‘were’, and not ‘was’
Figure 10 rates(%) – add a space
Line 394 fungi or fungus?
Line 403- ‘promot’ – rather promote ??
Line 418-419 in the cited paper [39] there is no information about 'triohodexrmin, gliotoxin, viridin, and peptide antibiotics [39].' - please cite the appropriate paper!
Line 466 it should be fungus instead of fungi
